# The First Heterozygous *TWNK* Nonsense Mutation Associated with Progressive External Ophthalmoplegia: Evidence for a New Piece in the Puzzle of Mitochondrial Diseases

**DOI:** 10.3390/biom15091337

**Published:** 2025-09-18

**Authors:** Diego Lopergolo, Gianna Berti, Gian Nicola Gallus, Silvia Bianchi, Filippo Maria Santorelli, Alessandro Malandrini, Nicola De Stefano

**Affiliations:** 1Department of Medicine, Surgery and Neurosciences, University of Siena, 53100 Siena, Italy; 2UOC Neurologia, Azienda Ospedaliero-Universitaria Senese, 53100 Siena, Italy; 3Neurobiology and Molecular Medicine, IRCCS Fondazione Stella Maris, 56128 Pisa, Italy

**Keywords:** *TWINKLE*, Twinkle-related disorders, myopathy, mitochondrial disease

## Abstract

Background: The *TWNK* gene encodes a protein that colocalizes with mitochondrial DNA (mtDNA) in mitochondrial nucleoids. It acts as mtDNA helicase during replication, thus playing a pivotal role in the replication and maintenance of mtDNA stability. *TWNK* mutations are associated with a wide spectrum of clinical phenotypes and a marked heterogeneity. However, heterozygous nonsense variants in the gene have never been described in association with disease. Methods: We analyzed a next-generation sequencing (NGS) targeted gene panel in a cohort including 40 patients with high clinical suspicion of mitochondrial disorders. Selected patients underwent a complete neurological examination, electrophysiology tests, and muscle biopsy. Segregation analysis was performed in available family members. The 3D structure of twinkle was visualized and analyzed using Swiss Model and Pymol version 3.1.6.1. Results: We found four *TWNK*-mutated subjects from two unrelated families. They exhibited a variable clinical spectrum, ranging from asymptomatic individuals to subjects with psychiatric disorder, chronic progressive external ophthalmoplegia (CPEO), and CPEO-plus. All the subjects shared the heterozygous *TWNK* p.Glu665Ter variant. Discussion and Conclusions: We describe the clinical phenotype and muscle biopsy findings associated with the first reported heterozygous nonsense *TWNK* variant, thus expanding the current knowledge of Twinkle-related disorders. Our findings are in line with the high intrafamilial clinical variability associated with *TWNK* mutations. Although PEO and skeletal muscle involvement remain hallmarks of the disease, extra-muscular features should be carefully assessed.

## 1. Introduction

Mitochondria are dynamic cellular organelles responsible for producing the majority of cellular ATP through processes of oxidative phosphorylation. Each mitochondrion contains several copies of mitochondrial DNA (mtDNA), a circular double-stranded molecule of 16.6 kb that encodes 13 essential subunits of the respiratory chain, along with 22 tRNAs and 2 rRNAs [1]. Proper replication and maintenance of mtDNA are required and they represent critical processes for mitochondrial function and cellular energy homeostasis. Indeed, defects in mtDNA replication often result in multisystem disorders with heterogeneous clinical presentations, especially affecting organs and tissues with high energy demands such as the brain, skeletal muscle, and heart [2].

The *TWNK* gene encodes Twinkle, a 684-amino acid mitochondrial DNA helicase that shares homology with the phage T7 gp4 helicase/primase and localizes within mitochondrial nucleoids [3]. Twinkle plays an essential role in mtDNA replication by unwinding the double-stranded DNA in a 5′ to 3′ direction ahead of the replication fork [4]. Twinkle operates in concert with other key replication factors, including DNA polymerase gamma (POLG) and mitochondrial single-stranded DNA-binding protein (mtSSB), forming the mitochondrial replisome [5]. Functional studies have shown that mutations in *TWNK* can impair helicase activity, disrupt mtDNA replication, and lead to mtDNA deletions or depletion, ultimately compromising respiratory chain function [6].

Pathogenic *TWNK* variants are associated with a broad spectrum of mitochondrial diseases. These include autosomal dominant progressive external ophthalmoplegia (PEO), characterized by bilateral ptosis, external ophthalmoplegia, and skeletal muscle weakness [2]; infantile-onset spinocerebellar ataxia (IOSCA), a severe autosomal recessive neurodegenerative condition [7]; and mitochondrial DNA depletion syndromes (MDDSs), a group of autosomal recessive disorders which present in early childhood with encephalopathy, liver dysfunction, and failure to thrive [8]. Most disease-causing *TWNK* variants reported to date are missense mutations affecting highly conserved regions within the helicase domain [2,9]. These mutations are thought to produce dominant-negative or loss-of-function effects, resulting in defective mtDNA maintenance.

Despite increasing recognition of Twinkle-related disorders, the mutational spectrum of the gene remains incompletely characterized. Notably, to date, no nonsense heterozygous *TWNK* variants have been reported in association with autosomal dominant disease. This represents a significant gap in our understanding of the gene’s tolerance to truncating variants and the phenotypic consequences of such mutations. *TWNK* nonsense variants have in fact previously been reported only in association with autosomal recessive disease [10,11,12].

Nonsense variants are present in gnomAD v4.1.0, although loss of function variants are under-represented indicating some evidence for evolutionary constraint [13], raising questions about their viability or potential pathogenic mechanisms.

In this study, we describe four individuals from two unrelated families carrying the same heterozygous variant in *TWNK* (p.Glu665Ter), representing the first report of a heterozygous truncating mutation in this gene associated with disease. Through detailed clinical, genetic, and histopathological evaluation, we explore the phenotypic variability associated with this novel mutation, ranging from asymptomatic carriers to individuals with chronic progressive external ophthalmoplegia (CPEO) and CPEO-plus phenotypes. Our findings broaden the mutational spectrum of *TWNK* and provide new insights into the genotype–phenotype correlations in Twinkle-related mitochondrial diseases.

## 2. Materials and Methods

We analyzed a next-generation sequencing (NGS) panel of genes associated with mitochondrial disorders in a cohort of 40 patients with a high clinical suspicion. Selected patients underwent further evaluations, including neurological examination, family history collection, electromyography (EMG), electroneurography (ENG), and muscle biopsy. Segregation analysis was performed in available family members. Detailed clinical data, including age of onset, distribution of weakness, respiratory status, and cardiac involvement, were collected. Several follow-up visits were conducted to provide a longitudinal assessment of selected patients.

### 2.1. Standard Protocol Approvals and Patient Consents

All subjects involved in this study provided written informed consent. All experiments described were approved by the local ethical committee (Regional Ethics Committee for Clinical Trials of the Tuscany Region, n. 17397, approval date 20 July 2020). All procedures were conducted in accordance with the Helsinki Declaration of 1975.

### 2.2. Morphological Analysis of Muscle Biopsies

Routine morphology and immunohistochemical analyses of muscle proteins were performed on biopsy samples of the right or left rectus femoris muscle according to standard procedures [14]. Ten-micrometer-thick sections were stained using standardized histological and histochemical methods including hematoxylin and eosin, reduced NADH, SDH, and COX.

### 2.3. DNA Analysis

Total genomic DNA was obtained from peripheral blood samples with a MasterPure Complete DNA Purification Kit (Epicentre MasterPure DNA Purification Kit, cat#: MCD85201, Biosearch Technologies, Hoddesdon, UK). The analysis was performed though the whole-exome sequencing (WES) technique with NovaSeq6000 (Illumina, San Diego, CA, USA). A targeted panel including 241 genes associated with mitochondrial disorders including all known genes associated with PEO with multiple mitochondrial DNA deletions was analyzed in detail. Mutational analysis was carried on using GATK version v.4.0. External datasets (1000 genomes, ExAC, GnomAD) were used to study unidentified novel variants. A prioritization of the variants allowed us to select frameshift, splice, stopgain or stoploss variants, missense variants predicted to be damaging by CADD-phred v1.7 prediction tools, and variants with minor allele frequency (MAF) < 0.01. Sanger sequencing was performed to confirm selected variants. The American College of Medical Genetics and Genomics (ACMG) criteria were used for variant classification.

MtDNA extracted from muscle tissue was analyzed for deletions by long range- polymerase chain reaction (PCR) and Southern blotting as previously described by our group [15,16].

Long-range PCR assay was performed using primers (100F-8600R and 7400F-15000R) located in specific regions of the mitochondrial genome using the Expand Long Template PCR System (Roche Applied Science, Mannheim, Germany).

DNA was digested with Pvu II prior to Southern blotting and hybridized to a mtDNA probe generated with the Expand Long Template PCR System (Roche, Mannheim, Germany). mtDNA fragments were amplified by PCR and PCR products were then purified with QIAquick PCR Purification kits (Qiagen, Hilden, Germany) and analyzed. The DNA probe was labeled with either chemiluminesence or radioactivity and detected by autoradiography.

The sequencing of the entire mtDNA genome were also performed as previously described [17]; sequence data were analyzed using Chromas software version 2.2.0 and compared with the revised Cambridge reference sequence [18]. All substitutions were confirmed by PCR/restriction fragment length polymorphism analysis. Densitometric analysis was performed to estimate the amount of mutant mtDNA.

### 2.4. Three-Dimensional (3D) Modeling

Three-dimensional modeling was conducted using SwissModel (“https://swissmodel.expasy.org/” accessed on 12 July 2025) to show the potential impact of the found variant on the Twinkle protein structure. The analysis utilized the near-atomic resolution structure of the highly similar (87.12%) Q8CIW5.1.A template from Mus musculus (GMQE = 0.79). The 3D structures of wild-type and mutated protein were also generated using human twinkle deposited in the Protein Data Bank and visualized using PyMol (PyMOL v. 3.1.6.1).

## 3. Results

### 3.1. Clinical Description

#### 3.1.1. Family 1

Subject 1 is a 59-year-old woman. Her family history is notable for her mother’s death at age 27 due to pregnancy-induced hypertension and her father’s death at age 54 from ischemic heart disease after suffering a myocardial infarction at 46 years. Maternal aunts died at old age without neurological disorders, and two sisters, sharing the same father, appeared in good health (Figure 1a).

Subject 1 first experienced ptosis and photophobia at age 35, followed by gait instability. Later, she developed anxiety and depressive symptoms. At approximately 45 years of age, the subject reported a sensation of heaviness in the shoulders and facial muscle tension, alongside progressive worsening of gait instability and imbalance. Brain MRI at this time showed several white matter signal alterations on long TR sequences, in the semioval centrum, frontal, and peritrigonal white matter, accompanied by mild enlargement of the periencephalic spaces. Routine laboratory tests revealed hyperkalemia (6.1 mEq/L), elevated magnesium (2.60 mg/dL), mild CPK elevation (176 U/L), and leukocytosis. Thyroid function tests demonstrated elevated free T3 levels (9.3 pg/mL). Serum lactate and pyruvate levels were elevated: 2 (reference range 0.3–1.3 mmol/L) and 0.12 mmol/L (reference range 0.03–0.08 mmol/L), respectively, with a decreased lactate/pyruvate ratio (17, reference range 10–20).

At the age of 48 years, the subject exhibited mild worsening of gait disturbances and mild dysphagia. Neurological exam showed marked ptosis, ophthalmoplegia, broad-based and slightly steppage gait, inability to walk on heels, and impaired directional changes. Deep tendon reflexes were absent except for brisk patellar reflexes. Therapy with an SSRI (Citalopram 20 mg/day) was initiated for anxiety.

By age 50, progressive deterioration of gait and daily functioning was evident. She experienced severe anxiety requiring alprazolam treatment. Neurological assessment revealed ophthalmoplegia, occasional dysphagia, dysarthria, mild limb (MRC = 4/5) and neck (MRC = 3/5) weakness, and ataxic gait. Treatment with liquid ubidecarenone (CoQ10) and carnitine supplementation was started. No significant improvement was noted after one year of liquid ubidecarenone; therefore, idebenone, a synthetic CoQ10 analog with better blood–brain barrier penetration, was introduced alongside carnitine. The subject reported slight relief of muscle contractures with this therapeutic change.

At age 52, neurological examination confirmed ophthalmoplegia, occasional dysphagia, dysarthria, mild weakness of all four limbs and neck muscles, ataxic gait with mild steppage, brisk upper limb reflexes, absent reflexes in some lower limb muscles, and mild dysmetria. Anxiety and depressive symptoms persisted.

At 53 and 54 years, progressive worsening of ataxia was noted, with increased gait instability requiring unilateral support. Complete ptosis required daily use of eyelid tape, which caused local irritation. CK levels markedly increased (1983 U/L). ECG remained within normal limits. Brain MRI revealed white matter abnormalities and generalized cerebral atrophy.

At 59 years, chest CT demonstrated ground-glass opacities consistent with interstitial lung disease. Muscle enzyme levels remained elevated (451 U/L). Cardiac evaluation showed bi-atrial hypertrophy. Neurological examination remained consistent with prior findings, including mild sensorineural hearing loss and mild dysarthria. Audiometric testing confirmed sensorineural hearing loss. Speech therapy was recommended to assist with dysphagia management.

Subject 2 is a 50-year-old-female, the sister of subject 1. She was asymptomatic and a neurological examination performed at 44 years old was normal. CPK dosage was within normal limits. Serum lactate and pyruvate levels were 0.7 (reference range 0.3–1.3 mmol/L) and 0.03 mmol/L (reference range 0.03–0.08 mmol/L), respectively. The lactate/pyruvate ratio was mildly elevated (23). The neurophysiological study by ENG/EMG was normal, excluding widespread involvement of peripheral nervous and muscular systems.

#### 3.1.2. Family 2

Subject 3 is a 40-year-old woman. Family history revealed an 8-year-old daughter with easy fatigability and muscle pain, a brother deceased in the perinatal period, a 47-year-old sister in apparent good health, a 73-year-old father, a mother deceased at 56 years from acute leukemia, with depression and respiratory disorders, a maternal grandmother deceased at 92 years with a reported history of psychiatric disorders, and a maternal grandfather deceased at 70 years due to lung cancer (Figure 1b).

At the age of 27, during her first pregnancy, the subject presented with fatigue, respiratory difficulties during swallowing and conversation with a sensation of air hunger, and diffused muscle cramps. Due to persistence of these symptoms, the subject underwent initial investigations at approximately 30 years of age: anti-AChR and anti-MuSK antibody titers, vitamin D levels, thyroid function tests, immunoglobulin A, G, and M levels, EMG, ENG, and brain MRI were all within normal limits. The subject also underwent an ischemic test with lactate and ammonia curve analysis, which was normal.

Neurological examinations at 32 years showed no strength deficits in the upper limbs, difficulty maintaining the Mingazzini position in the lower limbs, normal cerebellar tests, normal gait, and brisk, symmetric deep tendon reflexes in all four limbs. During the 6 min walk test, the subject covered 540 m, stopping once during the test, with a Borg scale score of 4 for both dyspnea and leg fatigue.

The subject underwent multiple pulmonary function tests and plethysmography, documenting a marked reduction in maximal expiratory pressure (MEP) and maximal inspiratory pressure (MIP). Respiratory volume and MIP values in the upright position progressively decreased with repeated testing, a phenomenon that disappeared in the supine position, where good reproducibility was observed. Chest CT scan performed at 32 years was normal; cardiopulmonary exercise testing showed early achievement of the anaerobic threshold and modest exercise tolerance. Echocardiography, ECG, and chest MRI were normal. Routine laboratory tests showed a mild increase in total bilirubin (1.4 mg/dL) and direct bilirubin (0.5 mg/dL).

At 33 years, the subject showed ptosis and dysphagia; around 35 years, onset of diplopia was reported.

Subject 4 is the sister of subject 3. She was asymptomatic, and neurological examination at 47 years old was normal. CPK dosage was within normal limits, and the neurophysiological study by ENG/EMG was normal.

### 3.2. Histopathological Features

Muscle biopsy performed in subject 1 at age 43 showed mild fiber size variability without necrosis, physiological nuclear internalization, and the presence of ragged red fibers on trichrome staining. Oxidative enzyme histochemistry revealed 35–40% of fibers negative for cytochrome c oxidase, highly consistent with mitochondrial myopathy (Figure 2a–c). The muscle biopsy of subject 2 was normal.

Muscle biopsy from subject 3 performed at age 32 showed size variability with the presence of some hypotrophic and hypertrophic fibers, nuclear internalizations within physiological percentage, absence of necrosis, normal oxidative reactions and acid phosphatase reaction, slight increase in sudanophilic lipids, and a low intensity of PAS reaction, with some fibers almost negative to the reaction (Figure 2d–f).

### 3.3. Genetic Analysis and 3D Structure Modeling

Molecular genetic testing in subjects 1 and 3 identified the novel heterozygous variant c.1992dup (p.Glu665Ter) in the TWNK gene (NM_021830.5), located in exon 5. Independently from the inheritance mechanism of TWNK gene, the variant should be classified as likely pathogenic according to the ACMG criteria (PVS1, PM2). However, because heterozygous TWNK null variants have not been identified as a known mechanism of pathogenicity consistent with autosomal dominant disease, the PVS1 criterion should not be applied in this specific clinical context [19]. The p.Glu665Ter variant results in the substitution of a glutamic acid residue at position 665 with a premature stop codon. This nonsense change predicts the synthesis of a truncated form of Twinkle, lacking the final 20 amino acids of the wild-type protein, which normally comprises 684 residues. Given that the premature termination codon occurs near the end of the final exon, the corresponding mRNA is unlikely to be targeted by nonsense-mediated mRNA decay (NMD), suggesting that the truncated protein is likely to be produced and accumulate in cells, albeit in a potentially dysfunctional form.

The identified variant was predicted to be damaging by the CADD-phred prediction tools (CADD-phred = 24.1). The minor allele frequency (MAF) was <0.01%, the ExAc all frequency of the variant was indetermined, and on gnomAD v4.1.0, the allele frequency is 0.00001177 (19/1614088 alleles, including 18/64010 Finnish alleles). The variant was Sanger-confirmed in subjects 1 and 3 (Figure 3a). Variant segregation analysis was performed in subject 1’s sister and subject 3’s sister. They both harbored the same variant in heterozygous state. Both of subject 1’s parents were deceased and not testable. However, since the variant was present in subjects 1 and 2, sharing the same father but having different mothers, the variant was likely inherited from their father. The variant was absent in subject 3’s father. Subject 3’s mother was deceased and not testable.

Long-range PCR analysis of skeletal muscle DNA and subsequent Southern blotting showed the presence of multiple mtDNA deletions in both subjects 1 and 3 (Figure 3b,c).

Whole mitochondrial DNA sequencing of muscle DNA in subjects 1 and 3 was performed to rule out a primary mitochondrial DNA cause. The analysis was normal in both subjects.

The 3D analysis demonstrated that the mutation, located in the C-terminal domain, leads to the production of a shorter protein (Figure 4c–f), although the single monomer structure seems to be preserved. However, the C-terminal domain of Twinkle is essential for the oligomerization into a functional hexamer [20], a conformation required for helicase activity, and supports interactions with mitochondrial replication factors. Therefore, truncation of this region could potentially impair the protein’s ability to unwind DNA or form a stable hexameric complex. This may obviously depend on the ratio of mutated and wild-type monomers involved in the composition of each hexameric complex. Our prediction indicates a likely stable hexameric complex when all the monomers are mutated (Figure 4a,b).

## 4. Discussion

In this study, we describe two families sharing the same heterozygous nonsense variant in *TWNK*, representing the first report of a heterozygous truncating change in this gene associated with disease. The phenotypic variability associated with this novel pathogenic variant ranges from asymptomatic carriers (subjects 2 and 4) to individuals with CPEO (subject 3) and CPEO-plus phenotypes (subject 1). Our findings broaden the mutational spectrum of *TWNK* and provide new insights into genotype–phenotype correlations.

We observed a high intrafamilial clinical variability associated with *TWNK* mutations. Although CPEO and skeletal muscle involvement remain hallmarks of the disease, extra-muscular features should be carefully assessed. The heterogeneity of clinical manifestations, even among carriers of the same pathogenic variant, suggests that genotype alone is not sufficient to predict phenotype, highlighting the complexity of Twinkle-related disorders [21,22].

Genotype–phenotype correlation remains a significant challenge in the interpretation of *TWNK* variants. Dominant mutations are typically associated with adult-onset PEO and multiple mtDNA deletions [21,22], while recessive mutations often lead to more severe and early-onset phenotypes [7,23]. However, increasing reports of intermediate or overlapping phenotypes blur these distinctions. Some dominantly inherited variants, such as p.Arg374Gln or p.Phe343Leu, have been found in patients presenting with both classical PEO and multisystemic involvement, including respiratory failure, dysphagia, and neuropsychiatric symptoms [21]. Conversely, recessive mutations such as p.Asn441Ser or p.Gly575Arg have shown incomplete penetrance or milder phenotypes in compound heterozygous states [7,22]. These findings suggest that the functional impact of each mutation on the helicase domain, mtDNA replication, or interaction with mitochondrial interactors (e.g., POLG, mtSSB) may vary substantially and influence the clinical course [24].

In our subjects, the presence of bulbar symptoms, respiratory dysfunction, and early fatigue in the absence of full-blown PEO in some individuals may reflect subtle impairment of mitochondrial dynamics rather than overt mtDNA instability. Such findings are consistent with recent in vitro studies showing that certain *TWNK* variants may impair helicase activity or mtDNA maintenance only under conditions of cellular stress, potentially explaining why phenotypic expression can be highly variable, even within the same family [24]. Moreover, modifying factors, including nuclear background, mitochondrial haplogroups, and environmental influences (e.g., hormonal changes during pregnancy, as observed in our patients), may further contribute to clinical heterogeneity [25]. The presence of psychiatric symptoms in multiple family members also raises the question of whether mitochondrial dysfunction in specific brain regions might contribute to these non-motor manifestations, as suggested by growing evidence in the literature [25]. Accordingly, a different level of tissue involvement within the same individual has been previously suggested. Deschauer et al. [26] described a *TWNK*-mutated patient with PEO showing a very low level (3%) of COX-deficient fibers on muscle biopsy. Similarly, a patient with sporadic PEO harboring the pathogenic R627W *POLG* variant showed normal results from quadriceps muscle biopsy and Southern blotting [27]. In line with these cases showing little clinical evidence of peripheral muscle involvement, the rectus femoris muscle biopsy in our subject 3 was only mildly abnormal. However, Deschauer et al. [26] observed a percentage of COX-deficient cells in extraocular muscle significantly greater compared to quadriceps muscle from patients with single large-scale mtDNA rearrangements. Therefore, it would not be surprising to detect significant mitochondrial abnormalities in respiratory muscle tissue in subject 3, who has a prominent involvement of respiratory muscles. These findings further confirm the phenotypic expression of mitochondrial disease, notoriously variable even in the presence of the same genetic defect [28].

While population allele frequencies are lower for well-known pathogenic variants associated with dominant PEO, the allele frequency in the Finnish population in gnomAD is surprisingly high for our variant. This data seems to be consistent with the reduced penetrance observed in our families and with the hypothesis that our variant may be associated only with a subtle impairment of mitochondrial dynamics.

Other possible contributions to the high clinical heterogeneity associated with *TWNK* variants derive from the Twinkle protein structure. Twinkle is a multidomain protein whose helicase function depends on the structural and functional integrity of the N Terminal Domain (NTD), linker helix, and C Terminal Domain (CTD). The NTD (residues 54–214) constitutes approximately 50% of the protein and it facilitates DNA binding and oligomer stability [29]. Many pathogenic mutations are clustered at the interface between the NTD, the linker region, and the CTD, supporting the importance of inter-domain communication for helicase function [20,30]. The linker helix (residues 360–394) connects the NTD to the CTD and mediates inter-subunit coordination, which is essential for proper hexameric or heptameric ring formation [3,31]. The linker helix transmits ATP-driven conformational changes across subunits, allowing coordinated helicase motion [31]. Several pathogenic variants in this region destabilize ring formation or interfere with DNA loading, reinforcing its importance for helicase activity and structural integrity [20]. The CTD (between 394 and 684 residues) contains conserved helicase motifs and is directly responsible for ATP hydrolysis and 5′→3′ DNA unwinding [3,32].

Cryo-EM structural studies show TWINKLE assembles into a two-tiered hexamer, with the CTDs forming a lower ring and the NTDs forming a flexible upper ring [33]. Variants that disrupt the NTD or linker region frequently impair oligomerization and reduce helicase activity, even when ATPase function remains intact. Dominant mutations in these domains are commonly associated with PEO, while biallelic mutations can lead to early-onset spinocerebellar ataxia or mitochondrial DNA depletion syndromes [2,34]. Notably, truncated constructs lacking the NTD demonstrate reduced unwinding activity despite preserved ATP hydrolysis, underscoring the NTD’s non-enzymatic but essential role in DNA engagement and oligomer stability [35].

The coordinated action of all three domains—NTD, linker, and CTD—is necessary to ensure high-fidelity mtDNA replication. TWINKLE also interacts functionally with POLG and mtSSB, forming the minimal mitochondrial replisome [20,29,31,36].

Our TWNK variant is not predicted to be a null variant. Indeed, the premature termination codon occurs near the end of the final exon; thus, the corresponding mRNA is unlikely to be targeted by NMD and a stable truncated protein is predicted to be produced and accumulated in cells. It may thus impair the function of Twinkle hexamers/multimers, enabling the possibility of autosomal dominant disease. Our nonsense variant affects the C-terminal domain of Twinkle, essential for several critical functions. It facilitates oligomerization into a functional hexamer, a conformation required for helicase activity, and supports interactions with mitochondrial replication factors, including POLG and mtSSB [3,29]. Truncation of this region is therefore likely to impair both structural integrity and protein–protein interactions. Loss of the terminal 20 residues could disrupt one or more of these functions, potentially impairing the protein’s ability to unwind DNA or form a stable hexameric complex.

Additionally, proteins truncated near the C-terminus are often destabilized, prone to misfolding, and may be subject to degradation via proteasomal pathways [37]. In particular, the region between residues 640 and 684 seems to be involved in negative regulation of ATPase activity [38]. Therefore, truncation of this region could potentially impair the protein’s ability to unwind DNA or form a stable hexameric complex.

Since Twinkle acts as a stable hexameric complex, the mutated and wild-type ratio of monomers involved in the composition of each complex may influence the final protein activity. The high variability observed in our mutated patients may thus also be derived from a different ratio of mutated monomers in the final hexameric complex. Moreover, the relatively high frequency of heterozygous TWNK truncating variants in reference databases may suggest a lower impact of truncating variants compared to missense variants to the final complex stability. Further studies, including a larger number of individuals harboring nonsense heterozygous TWNK variants, are needed to overcome the limitations of our study, including a small cohort of patients. Moreover, further evidence deriving from in vitro functional studies should be recommended to further investigate and support the findings and hypothesis of our study, in particular regarding the effect of the variant on protein oligomerization.

The limited predictive value of genotype alone highlights the need for integrative diagnostic strategies that combine clinical, biochemical, and functional analyses with molecular data. Muscle biopsy findings may support the diagnosis even in genetically unresolved or borderline cases. Additionally, longitudinal monitoring is crucial, as symptoms may evolve over time or emerge under stressors like infection, pregnancy, or aging.

## 5. Conclusions

We describe for the first time the clinical phenotype and muscle biopsy findings associated with a heterozygous nonsense *TWNK* variant, thus expanding our current knowledge of Twinkle-related disorders. Our findings are in line with the high intrafamilial clinical variability associated with TWNK mutations. Although PEO and skeletal muscle involvement remain hallmarks of the disease, extra-muscular features should be carefully assessed.

In conclusion, while *TWNK* mutations are strongly associated with mitochondrial myopathy and PEO, their phenotypic expression is modulated by a complex interplay of genetic and non-genetic factors. Future studies integrating large-scale genotype–phenotype correlations, functional assays, and omics approaches are essential to better understand disease mechanisms and to refine prognostic and therapeutic strategies.

## Figures and Tables

**Figure 1 biomolecules-15-01337-f001:**
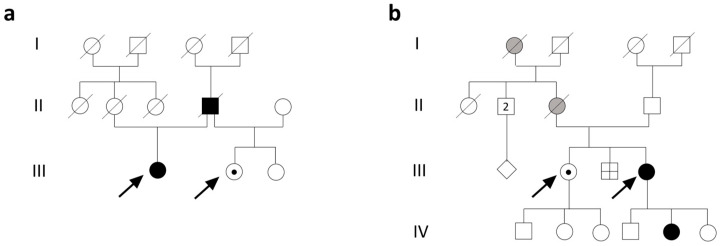
Pedigrees of family 1 (**a**) and 2 (**b**). (**a**) Subject 1 (III;1) is a 59-year-old woman; her mother (II;2) died at age 27 due to pregnancy-induced hypertension and her father (II;4) died at age 54 from ischemic heart disease. Subject 2 (III;2) is a 50-year-old female. (**b**) Subject 3 (III;4) is a 40-year-old female. Her 8-year-old daughter (IV;5) had easy fatigability and muscle pain; her brother (III;3) died in the perinatal period; her 47-year-old sister is in apparent good health (subject 4, III;2); her mother (II;3) died at 56 years from acute leukemia, with depression and respiratory disorders; and her maternal grandmother (I;1) died at 92 years with a reported history of psychiatric disorders. Key symbols include squares for males, circles for females, and diamonds for individuals with unknown sex or gender; shaded symbols indicate affected (black) or mildly affected (gray) individuals; a symbol with a line through indicates a deceased individual; a symbol with a black point indicates unaffected heterozygous carriers.

**Figure 2 biomolecules-15-01337-f002:**
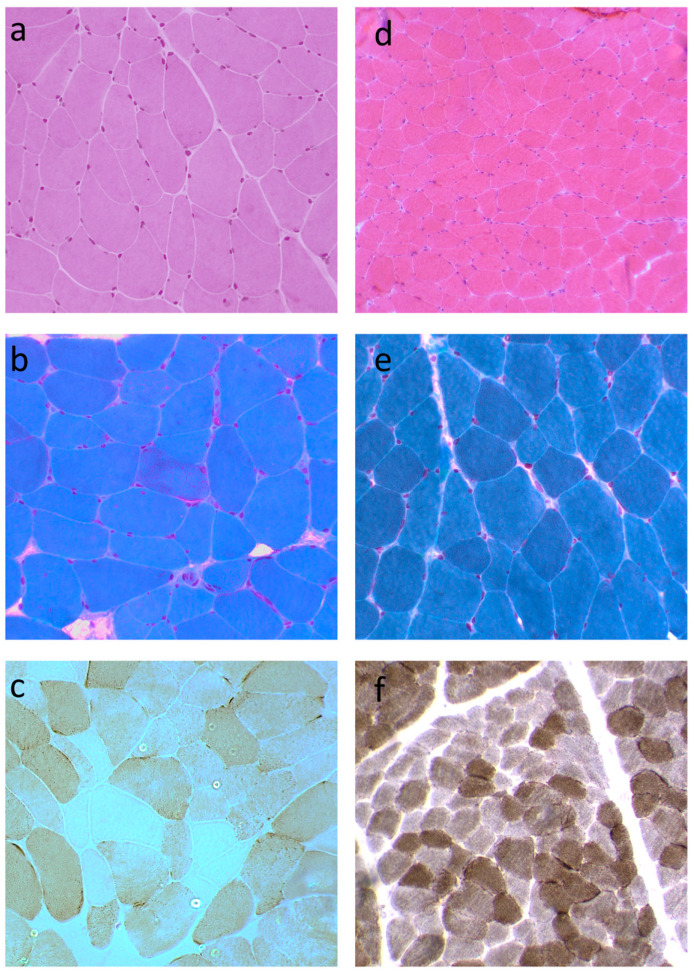
Muscle biopsy of subjects 1 and 3. (**a**) Hematoxylin-and-eosin staining in subject 1 showed mild fiber size variability (×20). (**b**) A ragged red fiber in subject 1 is evident in the center of picture on trichrome staining (×20). (**c**) Cytochrome c oxidase staining in subject 1 shows fibers negative to the reaction (×20). (**d**) Hematoxylin-and-eosin staining in subject 3 showed fiber size variability (×10). Trichrome (×20) (**e**) and cytochrome c oxidase staining (×10) (**f**) were essentially normal in subject 3.

**Figure 3 biomolecules-15-01337-f003:**
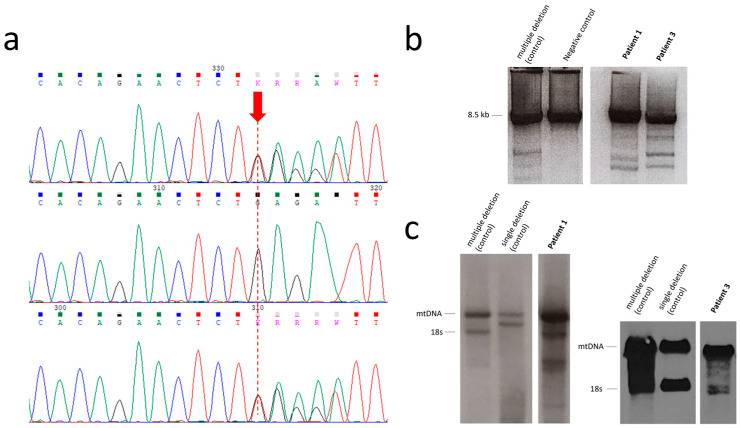
(**a**) The heterozygous *TWNK* variant c.1992dup (p.Glu665Ter). Top line, Sanger sequencing of subject 3 (variant present); middle line, the father of subject 3 (variant absent); bottom line, subject 1 (variant present). The variant’s location is indicated with a red arrow. (**b**) Long-range PCR of muscle samples from subjects 1 and 3. Multiple mtDNA deletions are evident as multiple bands in the corresponding lanes. (**c**) Southern blotting. The analysis confirmed the presence of multiple deletions in the subjects’ samples, evident as multiple bands corresponding to partially deleted mitochondrial genomes. Original Western blot images can be found in Appendix A.

**Figure 4 biomolecules-15-01337-f004:**
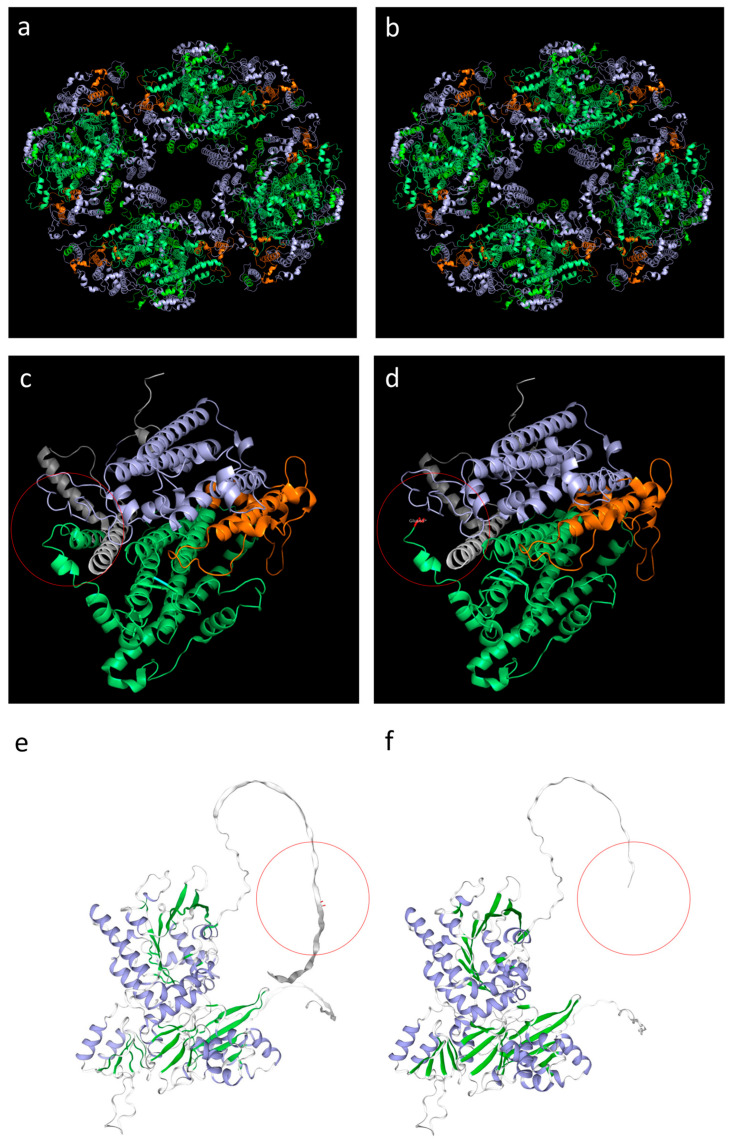
Three-dimensional analysis of the twinkle mutation. Wild-type (**a**) and mutated (**b**) Twinkle oligomerization into a hexamer protein. Wild-type (**c**,**e**) and mutated (**d**,**f**) Twinkle monomers: mutation site is indicated by red circles. The loss of the terminal 20 residues due to the nonsense variant likely impairs both structural integrity and protein–protein interactions, potentially impairing the protein’s ability to form a stable hexameric complex. C-terminal domains are indicated in green, N-terminal domains in gray, linker helixes in orange (**a**–**d**).

## Data Availability

The data that support the findings of this study are available from the corresponding author upon reasonable request.

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
