# Peer review of "The First Heterozygous TWNK Nonsense Mutation Associated with Progressive External Ophthalmoplegia: Evidence for a New Piece in the Puzzle of Mitochondrial Diseases"

_biomolecules, 2025, doi:10.3390/biom15091337_

Round 1

Reviewer 1 Report

Comments and Suggestions for Authors

This manuscript by Lopergolo et al. reports the first nonsense variant in the TWNK gene (p.Glu665Ter), identified in two unrelated families. The study is well conducted and adds useful data to the field of mitochondrial genetics by expanding the mutational spectrum of TWNK and showing the variability associated with truncating variants.

The paper is clearly written and structured. The clinical descriptions are detailed, the histopathological and genetic analyses are appropriate, and the modeling data are informative. The authors are also careful in their interpretation.

Overall, this is a solid descriptive paper that fits the scope of Biomolecules. I only have a few minor suggestions:

1. Use terminology consistently: either "Twinkle-related disorders" or "TWNK-related disorders" throughout.

2. p.5 row 208: “Sisters” should be “Sister”.

3. p.6 row 235: A stop codon at position Glu665 generates a protein lacking 20 amino acids. Please note that Glu seems not included in the calculation.

4. p.7 row 246: “Both patient 1’s parents were deceased and not testable…” should probably read “patients 1 and 2” instead of “patients 1 and 3”.

5. p.7 row 253–254: “the C-terminal domain of Twinkle is essential for the oligomerization into a functional hexamer”. Please add a reference. This statement comes from work on the truncated variant ‘Twinky’, which lacks residues 579–684 and therefore a large part of the C-terminal domain. The present mutation affects only the very C-terminal tail. In line with the authors’ prediction, our unpublished observations suggest that this tail is not important for hexamerization.

6. p.9 row 270–271: “pathogenic variant ranges from asymptomatic carriers (patients 2 and 4) to individuals with CPEO (patient 2)”. Please check if this is correct.

Author Response

Reviewer 1

This manuscript by Lopergolo et al. reports the first nonsense variant in the TWNK gene (p.Glu665Ter), identified in two unrelated families. The study is well conducted and adds useful data to the field of mitochondrial genetics by expanding the mutational spectrum of TWNK and showing the variability associated with truncating variants. The paper is clearly written and structured. The clinical descriptions are detailed, the histopathological and genetic analyses are appropriate, and the modeling data are informative. The authors are also careful in their interpretation.

Overall, this is a solid descriptive paper that fits the scope of Biomolecules. I only have a few minor suggestions:

1. Use terminology consistently: either "Twinkle-related disorders" or "TWNK-related disorders" throughout.

We apologize to the reviewer. We used consistent terminology in the revised manuscript.

 2. p.5 row 208: “Sisters” should be “Sister”.

We apologize. We have corrected the typo.

 3. p.6 row 235: A stop codon at position Glu665 generates a protein lacking 20 amino acids. Please note that Glu seems not included in the calculation.

We apologize. We have corrected the indicated number.

 4. p.7 row 246: “Both patient 1’s parents were deceased and not testable…” should probably read “patients 1 and 2” instead of “patients 1 and 3”.

We apologize. We have corrected the typo.

 5. p.7 row 253–254: “the C-terminal domain of Twinkle is essential for the oligomerization into a functional hexamer”. Please add a reference. This statement comes from work on the truncated variant ‘Twinky’, which lacks residues 579–684 and therefore a large part of the C-terminal domain. The present mutation affects only the very C-terminal tail. In line with the authors’ prediction, our unpublished observations suggest that this tail is not important for hexamerization.

We apologize and we thank the reviewer. We have added a reference to our sentence.

 6. p.9 row 270–271: “pathogenic variant ranges from asymptomatic carriers (patients 2 and 4) to individuals with CPEO (patient 2)”. Please check if this is correct.

We apologize. We have revised the phrase in the manuscript.

Reviewer 2 Report

Comments and Suggestions for Authors

The manuscript is clearly presented and well written. The authors report 2 unrelated individuals with evidence of PEO with multiple mitochondrial DNA deletions, both of whom were found to have a rare heterozygous nonsense variant in the TWNK gene, which appears consistent with the clinical phenotype and muscle biopsy findings. The authors state that this is the first report of association of a TWNK nonsense variant with disease (most reported disease causing TWNK variants being missense).

Main concerns:

  • TWNK nonsense variants have in fact previously been reported in association with disease, e.g. c.85C>T p.Arg29Ter by Goh et al. 2012 (PMID: 21681116) and Dominguez-Ruiz et al. 2019 (PMID: 31455392), c.649C>T p.Arg217Ter by Hu et al. 2018 (PMID: 29095814). There are also several nonsense variants classified on ClinVar as (likely) pathogenic. However, as far as I’m aware, only association with autosomal recessive (biallelic) disease has been reported previously in the literature; therefore, the putative association with autosomal dominant PEO in this study is still novel. The authors should update the manuscript to reflect this.
  • In my view, the TWNK nonsense variant in this study cannot be definitively classified as pathogenic in association with autosomal dominant disease and so the manuscript should report this variant as a putative causative variant in these patients. The authors state that they use the ACMG criteria to assess variant pathogenicity, but this does not appear to have been undertaken correctly; in relation to PVS1 criterion, Richards et al. 2015 (PMID: 25741868) state: “When classifying such variants as pathogenic, one must ensure that null variants are a known mechanism of pathogenicity consistent with the established inheritance pattern for the disease” and “One must also be cautious when interpreting truncating variants downstream of the most 3′ truncating variant established as pathogenic in the literature. This is especially true if the predicted stop codon occurs in the last exon or in the last 50 base pairs of the penultimate exon, such that nonsense-mediated decay would not be predicted, and there is a higher likelihood of an expressed protein.” To expand on the second point, the guidance provided by Tayoun et al. 2018 (PMID: 30192042) is widely adopted. In any case, application of PVS1 for this variant in the context of autosomal dominant disease does not appear appropriate. Therefore, the variant has to be classified as a variant of uncertain significance (VUS). Further evidence, such as in vitro functional studies, is recommended to further investigate/support the findings in this study. This is particularly important, given the lack of supportive co-segregation in the two families reported. Also, the allele frequency in the Finnish population in gnomAD is surprisingly high for a variant that has not previously been reported in association with disease (population allele frequencies are lower for well known pathogenic variants associated with dominant PEO); the authors should perhaps comment on this, which could be consistent with the reduced penetrance suggested by this study.
  • Further discussion of the clinical significance of loss of function variants in TWNK is required. It is important to highlight that this variant is not predicted to be a null variant, i.e. a stable protein is predicted to be produced. A true null variant (with no protein expressed) would be expected to be capable of causing autosomal recessive (biallelic) disease, but not dominant disease. However, the predicted truncated protein in this study may impair the function of Twinkle hexamers/multimers, enabling the possibility of autosomal dominant disease (which the authors have discussed to some extent).
  • Details of the targeted panel of genes associated with mitochondrial disorders have not been provided. As a minimum, the number of genes in the panel should be stated (ideally added to section 2.3) along with a statement that this panel included all other known genes associated with PEO with multiple mitochondrial DNA deletions. In addition, although evidence of multiple mtDNA deletions was reported in muscle from patients 1 and 3, for completeness whole mitochondrial DNA sequencing of muscle DNA should be undertaken, to rule out a primary mitochondrial DNA cause.
  • As evidence of multiple mtDNA deletions is important supporting evidence, it would be helpful to include the long range PCR results demonstrating multiple mtDNA deletions in muscle DNA from patients 1 and 3 in a figure (perhaps supplementary if the number of figures permitted in main article is limited).

Minor issues:

  • Introduction, page 2, lines 51 and 52: it would be helpful to state that IOSCA and MDDS are autosomal recessive disorders.
  • Introduction, page 2, line 61: nonsense variants are present in gnomAD v4.1.0, although loss of function variants are under-represented indicating some evidence for constraint. Therefore, the sentence should be updated.
  • Figure 1 legend, page 3, line 125: the mother of patient 1 appears to be II;2 in the pedigree, not II;1.
  • Results, page 4, line 141: I think the lactate and pyruvate values are swapped and units should be mmol/L.
  • Results, page 4, lines 173 & 174: this sentence should be modified to clearly state which result is lactate and which pyruvate.
  • Results, page 5, line 208: “sisters” should be “sister”.
  • Results, page 6, line 231: the TWNK reference transcript should be stated, i.e. NM_021830.5.
  • Results, page 7, line 242: on gnomAD v4.1.0, the allele frequency appears to be 0.00001177; please check. It would also be helpful to state, e.g. 19/1614088 alleles (including 18/64010 Finnish alleles).
  • Discussion, page 9, line 305: I think “variant” should be “variants”.

Author Response

Reviewer 2

The manuscript is clearly presented and well written. The authors report 2 unrelated individuals with evidence of PEO with multiple mitochondrial DNA deletions, both of whom were found to have a rare heterozygous nonsense variant in the TWNK gene, which appears consistent with the clinical phenotype and muscle biopsy findings. The authors state that this is the first report of association of a TWNK nonsense variant with disease (most reported disease causing TWNK variants being missense).

Main concerns:

  • TWNK nonsense variants have in fact previously been reported in association with disease, e.g. c.85C>T p.Arg29Ter by Goh et al. 2012 (PMID: 21681116) and Dominguez-Ruiz et al. 2019 (PMID: 31455392), c.649C>T p.Arg217Ter by Hu et al. 2018 (PMID: 29095814). There are also several nonsense variants classified on ClinVar as (likely) pathogenic. However, as far as I’m aware, only association with autosomal recessive (biallelic) disease has been reported previously in the literature; therefore, the putative association with autosomal dominant PEO in this study is still novel. The authors should update the manuscript to reflect this.

We thank the reviewer for his suggestion. We revised the manuscript accordingly.

  • In my view, the TWNK nonsense variant in this study cannot be definitively classified as pathogenic in association with autosomal dominant disease and so the manuscript should report this variant as a putative causative variant in these patients. The authors state that they use the ACMG criteria to assess variant pathogenicity, but this does not appear to have been undertaken correctly; in relation to PVS1 criterion, Richards et al. 2015 (PMID: 25741868) state: “When classifying such variants as pathogenic, one must ensure that null variants are a known mechanism of pathogenicity consistent with the established inheritance pattern for the disease” and “One must also be cautious when interpreting truncating variants downstream of the most 3′ truncating variant established as pathogenic in the literature. This is especially true if the predicted stop codon occurs in the last exon or in the last 50 base pairs of the penultimate exon, such that nonsense-mediated decay would not be predicted, and there is a higher likelihood of an expressed protein.” To expand on the second point, the guidance provided by Tayoun et al. 2018 (PMID: 30192042) is widely adopted. In any case, application of PVS1 for this variant in the context of autosomal dominant disease does not appear appropriate. Therefore, the variant has to be classified as a variant of uncertain significance (VUS). Further evidence, such as in vitro functional studies, is recommended to further investigate/support the findings in this study. This is particularly important, given the lack of supportive co-segregation in the two families reported. Also, the allele frequency in the Finnish population in gnomAD is surprisingly high for a variant that has not previously been reported in association with disease (population allele frequencies are lower for well known pathogenic variants associated with dominant PEO); the authors should perhaps comment on this, which could be consistent with the reduced penetrance suggested by this study.

We thank the reviewer for his suggestions. In the revised manuscript we better explain the ACMG interpretation of the variant according to the reviewer observation.

Because in vitro functional studies are not available in our laboratory, in the revised manuscript we have added a section including the limitations of the study.

We also added in the revised manuscript a comment about the allele frequency in the Finnish population.

  • Further discussion of the clinical significance of loss of function variants in TWNK is required. It is important to highlight that this variant is not predicted to be a null variant, i.e. a stable protein is predicted to be produced. A true null variant (with no protein expressed) would be expected to be capable of causing autosomal recessive (biallelic) disease, but not dominant disease. However, the predicted truncated protein in this study may impair the function of Twinkle hexamers/multimers, enabling the possibility of autosomal dominant disease (which the authors have discussed to some extent).

We thank the reviewer for his suggestions. In the revised manuscript we better explain the predicted effect of the variant.

  • Details of the targeted panel of genes associated with mitochondrial disorders have not been provided. As a minimum, the number of genes in the panel should be stated (ideally added to section 2.3) along with a statement that this panel included all other known genes associated with PEO with multiple mitochondrial DNA deletions. In addition, although evidence of multiple mtDNA deletions was reported in muscle from patients 1 and 3, for completeness whole mitochondrial DNA sequencing of muscle DNA should be undertaken, to rule out a primary mitochondrial DNA cause.

We revised our manuscript according to the reviewer’s suggestions. Moreover, we have added the results of whole mitochondrial DNA sequencing of muscle DNA from patients 1 and 3, that ruled out to be normal.

  • As evidence of multiple mtDNA deletions is important supporting evidence, it would be helpful to include the long range PCR results demonstrating multiple mtDNA deletions in muscle DNA from patients 1 and 3 in a figure (perhaps supplementary if the number of figures permitted in main article is limited).

We thank the reviewer for his suggestion. In the revised manuscript we reported the figure related to the results of long-range PCR and southern blotting (new fig. 3).

Minor issues:

  • Introduction, page 2, lines 51 and 52: it would be helpful to state that IOSCA and MDDS are autosomal recessive disorders.
  • Introduction, page 2, line 61: nonsense variants are present in gnomAD v4.1.0, although loss of function variants are under-represented indicating some evidence for constraint. Therefore, the sentence should be updated.
  • Figure 1 legend, page 3, line 125: the mother of patient 1 appears to be II;2 in the pedigree, not II;1.
  • Results, page 4, line 141: I think the lactate and pyruvate values are swapped and units should be mmol/L.
  • Results, page 4, lines 173 & 174: this sentence should be modified to clearly state which result is lactate and which pyruvate.
  • Results, page 5, line 208: “sisters” should be “sister”.
  • Results, page 6, line 231: the TWNK reference transcript should be stated, i.e. NM_021830.5.
  • Results, page 7, line 242: on gnomAD v4.1.0, the allele frequency appears to be 0.00001177; please check. It would also be helpful to state, e.g. 19/1614088 alleles (including 18/64010 Finnish alleles).
  • Discussion, page 9, line 305: I think “variant” should be “variants”.

We apologize. We thank the reviewer for his suggestion. We revised our manuscript accordingly.

Reviewer 3 Report

Comments and Suggestions for Authors

Diego Lopergolo and colleagues report on the heterozygous TWNK stop variant c.1992dup (p.Glu665Ter) found in two affected female individuals (patient 1, patient 3) from two different families. These two affected individuals belong to a group 40 patients with suspected mitochondrial disease that was investigated by exome sequencing. Muscle biopsy of one of the two affected individuals showed cytochrome c oxidase negative and ragged red fibres while this was normal in the other affected individual. Multiple deletions of the mitochondrial DNA were detected in DNA from muscle of both affected individuals by long-range PCR. This TWNK stop variant was also found in two clinically normal siblings of the two affected individuals. The authors conclude that the identified TWNK stop variant is causative for the disease of the two affected individuals.

This is the first report that postulates that a heterozygous stop variant in TWNK might be causing mitochondrial disease.

Despite this observation is potentially remarkable, there are numerous questions and concerns regarding the hypothesis that truncating TWNK variants can be disease-causing:

  1. The reference number of the TWNK variant has not been provided. I assume that the variant is referred to NM_021830.5. Please clarify.

  1. Please show the TWNK sequence variant details (IGV of next generation sequencing and Sanger confirmation).

  1. Histologic investigation of muscle in patient 3 showed normal histologic results concerning mitochondrial function. How do the authors explain this result?

  1. The identification of multiple deletions of the mitochondrial DNA from muscle is a crucial finding in the two affected individuals. However, these results have not been shown and the method has not been explained in detail.

  1. Long-range PCR of mitochondrial DNA can result in artifacts and should be confirmed by other methods like Southern blotting or especially next generation sequencing of the DNA from muscle (e.g. PMID: 40164291).

  1. Concerning the pathogenic relevance of multiple deletions of the mitochondrial DNA, it would also be important to quantify the amount of deleted versus normal mitochondrial DNA. Long-range PCR is a very sensitive technique and might overestimate the amount of deleted mitochondrial DNA. Other techniques (cf. point 5 above) can give at least an estimate of the amount of deleted mitochondrial DNA. If the proportion of deleted mitochondrial DNA is below 10% of the total mitochondrial DNA, it is questionable if this has pathogenic relevance.

  1. A major concern in this study is the aspect of incomplete penetrance and different clinical expressivity of the disease. I agree that incomplete penetrance and different clinical expressivity have already been reported for other dominant TWNK variants, however, this is novel for stop variants in TWNK. The number of two affected individuals (from whom one did not show histological mitochondrial changes) is not very convincing.

  1. The characteristics of the TWNK stop variant is purely investigated. In my view, this TWNK stop variant might be interesting since it is located in the last exon of the TWNK gene. Therefore, it is likely to escape the nonsense-mediated mRNA decay. Quantification of the mRNA (e.g. by targeted techniques or by RNA-seq) could quantify the amount of the mutated versus the normal TWNK transcript.

  1. When looking at the reference databases like gnomAD, there are numerous heterozygous truncating variants in the TWNK gene, even a very similar variant (https://gnomad.broadinstitute.org/variant/10-100993446-C-CT?dataset=gnomad_r4) occurs in 19 out of 1614088 alleles. This is a strong argument against pathogenic relevance of such truncation variants in TWNK. How do you explain the relatively high frequency of loss of function in TWNK (even in parts of the gene that escape nonsense-mediated decay)?

  1. The authors speculate that the truncation of TWNK affects the oligomerisation of the TWNK protein. If you could show any experimental evidence for this hypothesis it would significantly improve the quality of your work.

Minor points:

  1. Was there any diagnostics performed in the 8-year-old daughter of patient 3, reported to suffer from easy fatigue?

  1. Pedigrees (Figure 1): Provide legend for each of the used symbols.

  1. Lines 140-142: Values for lactate and pyruvate seem to be interchanged. No references are provided for lactate, pyruvate and the lactate/pyruvate ratio.

  1. Why are the clinically unaffected heterozygous carriers called "patients"?

  1. Line 231: "TWNK" should be written in italics.

Author Response

Reviewer 3

Diego Lopergolo and colleagues report on the heterozygous TWNK stop variant c.1992dup (p.Glu665Ter) found in two affected female individuals (patient 1, patient 3) from two different families. These two affected individuals belong to a group 40 patients with suspected mitochondrial disease that was investigated by exome sequencing. Muscle biopsy of one of the two affected individuals showed cytochrome c oxidase negative and ragged red fibres while this was normal in the other affected individual. Multiple deletions of the mitochondrial DNA were detected in DNA from muscle of both affected individuals by long-range PCR. This TWNK stop variant was also found in two clinically normal siblings of the two affected individuals. The authors conclude that the identified TWNK stop variant is causative for the disease of the two affected individuals.

 This is the first report that postulates that a heterozygous stop variant in TWNK might be causing mitochondrial disease. 

Despite this observation is potentially remarkable, there are numerous questions and concerns regarding the hypothesis that truncating TWNK variants can be disease-causing:

  1. The reference number of the TWNK variant has not been provided. I assume that the variant is referred to NM_021830.5. Please clarify.

We apologize. We thank the reviewer for his suggestion. We revised our manuscript accordingly.

  1. Please show the TWNK sequence variant details (IGV of next generation sequencing and Sanger confirmation).

We thank the reviewer for the suggestion. In the revised manuscript we reported a new figure including the Sanger confirmation of the variant (the new fig. 3). Unfortunately, because the NGS sequencing was performed some years ago, the IGV data have been automatically deleted by the NAS system and they are no longer available in our laboratory. However, the presence of the variant in both the sisters in both the families rules out the possibility of low-grade mosaicism for the variant.

  1. Histologic investigation of muscle in patient 3 showed normal histologic results concerning mitochondrial function. How do the authors explain this result?

We thank the reviewer for the opportunity to better explain this point. Heterozygous TWNK mutations are usually associated with high clinical variability and incomplete penetrance. Deschauer et al (PMID: 12921794) described a TWNK mutated patient showing a very low level (3%) of COX-deficient fibres on muscle biopsy. This observation is similar to a recently reported patient with sporadic PEO and the pathogenic R627W POLG mutation, but whose quadriceps muscle biopsy and Southern blot were entirely normal (PMID: 12565911). In both these patients, there was little clinical evidence of peripheral muscle involvement; under these circumstances, the major effect of the genetic defect is being observed in the extraocular muscles. Accordingly, Deschauer et al observed that the percentage of COX-deficient cells in extraocular muscle is significantly greater than that seen in quadriceps from patients with single large-scale mtDNA rearrangements. These findings confirm the phenotypic expression of mitochondrial disease, notoriously variable even in the presence of the identical genetic defect (e.g. A3243G mutation, PMID: 12849395). Since our patient showed a prominent involvement of respiratory muscles, according to the previous findings, it’s likely a major involvement of such muscular district. Routine muscle biopsy was routinely performed on the rectus femoris muscle thus probably underestimate signs of mitochondrial myopathy. We had better focus on this point in the revised manuscript.

  1. The identification of multiple deletions of the mitochondrial DNA from muscle is a crucial finding in the two affected individuals. However, these results have not been shown and the method has not been explained in detail.

We apologize. We revised our manuscript accordingly.

  1. Long-range PCR of mitochondrial DNA can result in artifacts and should be confirmed by other methods like Southern blotting or especially next generation sequencing of the DNA from muscle (e.g. PMID: 40164291).

We thank the reviewer. In the revised manuscript we describe the results of Southern blotting, that confirmed the findings of the long-range PCR.

  1. Concerning the pathogenic relevance of multiple deletions of the mitochondrial DNA, it would also be important to quantify the amount of deleted versus normal mitochondrial DNA. Long-range PCR is a very sensitive technique and might overestimate the amount of deleted mitochondrial DNA. Other techniques (cf. point 5 above) can give at least an estimate of the amount of deleted mitochondrial DNA. If the proportion of deleted mitochondrial DNA is below 10% of the total mitochondrial DNA, it is questionable if this has pathogenic relevance.

We thank the reviewer for allowing us to better explain this point. After the results of Southern blotting that we have added to the revised manuscript, the estimation of deleted mitochondrial DNA is very high (more than 10%). In these cases, as the reviewer suggested, quantification of the amount of deleted versus normal mitochondrial DNA does not provide further evidence.

  1. A major concern in this study is the aspect of incomplete penetrance and different clinical expressivity of the disease. I agree that incomplete penetrance and different clinical expressivity have already been reported for other dominant TWNK variants, however, this is novel for stop variants in TWNK. The number of two affected individuals (from whom one did not show histological mitochondrial changes) is not very convincing.

We thank the reviewer for allowing us to better explain this point. Unfortunately, other individuals from the two families were not available to complete further variant segregation analyses. However, other than the two affected patients and the two mutated sisters, we must consider also the mother of patient 3, obligatory carrier of the mutation, who deceased at 56 years with depression and respiratory disorders.

Dominant Twinkle-related disorders represent a very rare condition with high clinical variability; we tried to describe such variability by describing patients sharing the same novel gene variant. Consequently, the number of possible patients is very limited.

However, we agree with the reviewer about the limited number of mutated individuals included in our study, therefore we included a sentence about the limitation of the study in the revised manuscript.

  1. The characteristics of the TWNK stop variant is purely investigated. In my view, this TWNK stop variant might be interesting since it is located in the last exon of the TWNK gene. Therefore, it is likely to escape the nonsense-mediated mRNA decay. Quantification of the mRNA (e.g. by targeted techniques or by RNA-seq) could quantify the amount of the mutated versus the normal TWNK transcript.

We thank the reviewer for his suggestion. Unfortunately, we have not the opportunity to study the effect of nonsense-mediated mRNA decay because of the absence of available biological samples from the patients.

  1. When looking at the reference databases like gnomAD, there are numerous heterozygous truncating variants in the TWNK gene, even a very similar variant (https://gnomad.broadinstitute.org/variant/10-100993446-C-CT?dataset=gnomad_r4) occurs in 19 out of 1614088 alleles. This is a strong argument against pathogenic relevance of such truncation variants in TWNK. How do you explain the relatively high frequency of loss of function in TWNK (even in parts of the gene that escape nonsense-mediated decay)?

In the revised manuscript we had better discuss this point. These data seem to be consistent with the hypothesis that our variant, like other truncated variants in gnomAD, may be associated only with a subtle impairment of mitochondrial dynamics. Since Twinkle acts as a stable hexameric complex, the mutated and wild type ratio of monomers involved in the composition of each complex may influence the final protein activity. The high variability and lower penetrance observed in our mutated patients may thus derive also by a different ratio of mutated monomers in the final hexameric complex. The relative high frequency of heterozygous TWNK truncating variants in reference databases may suggest a lower impact of truncating variants compared to missense variants to the final complex stability.

  1. The authors speculate that the truncation of TWNK affects the oligomerisation of the TWNK protein. If you could show any experimental evidence for this hypothesis it would significantly improve the quality of your work.

We thank the reviewer for his suggestion. Unfortunately, we have not the opportunity to study the effect of the variant on protein oligomerization. However, in the revised manuscript we inserted a statement about the limitation of our study.

Minor points:

  1. Was there any diagnostics performed in the 8-year-old daughter of patient 3, reported to suffer from easy fatigue?

Unfortunately, at her parents' request, the child has not been subjected to genetic testing or further investigations.

  1. Pedigrees (Figure 1): Provide legend for each of the used symbols.

We have added a legend as suggested.

  1. Lines 140-142: Values for lactate and pyruvate seem to be interchanged. No references are provided for lactate, pyruvate and the lactate/pyruvate ratio.

We apologize; we revised our manuscript accordingly.

  1. Why are the clinically unaffected heterozygous carriers called "patients"?

We apologize; we revised our manuscript accordingly.

  1. Line 231: "TWNK" should be written in italics.

We apologize; we revised our manuscript accordingly.

Round 2

Reviewer 2 Report

Comments and Suggestions for Authors

Thank you to the authors for fully addressing the issues that I raised in my initial review. I now only have the following minor points.

Minor issues:

  • Materials and Methods, page 3, line 117: “variant” should be “variants”.
  • Materials and Methods, page 3, line 121: “variants” should be “variant”.
  • Materials and Methods, page 3, line 121: “southern” should be “Southern”. This should also be corrected throughout the manuscript (it is correct in some places but not others).
  • Materials and Methods, page 3 & 4, lines 129-138: This description of the Southern blotting process is not clear to me and appears to be split across 2 paragraphs. Please can this be reworded? Does some of the lines 135-138 paragraph refer to the whole mitochondrial genome sequencing methodology?
  • Results, page 8, line 297: figure 3b should be referred to in the text here (as well as figure 3c).
  • Results, page 8, figure 3b: 2 lanes are marked as negative controls and appear to have different sized fragments; this should be corrected. Also, I think the positive control represents a positive control for multiple deletions; this should be clearly indicated (as has been done for figure 3c).
  • Discussion, page 10, line 372: “muscle biopsy in our subject 3 resulted only mildly affected” does not read well. It would be clearer to say “muscle biopsy in our subject 3 was only mildly abnormal.”
  • Discussion, page 10, lines 375-376: Again, this sentence is not clear. I suggest rewording along the following lines: “Therefore, it would not be surprising in subject 3, who has prominent involvement of respiratory muscles, to detect significant mitochondrial abnormalities in respiratory muscle tissue.”
  • Discussion, page 12, line 436: “TWKN” should be “TWNK”.

Author Response

REVIEWER 2

Thank you to the authors for fully addressing the issues that I raised in my initial review. I now only have the following minor points.

Minor issues:

    Materials and Methods, page 3, line 117: “variant” should be “variants”.

We thank the reviewer for his suggestion. We revised our manuscript accordingly.

    Materials and Methods, page 3, line 121: “variants” should be “variant”.

We thank the reviewer for his suggestion. We revised our manuscript accordingly.

    Materials and Methods, page 3, line 121: “southern” should be “Southern”. This should also be corrected throughout the manuscript (it is correct in some places but not others).

We thank the reviewer for his suggestion. We revised our manuscript accordingly.

    Materials and Methods, page 3 & 4, lines 129-138: This description of the Southern blotting process is not clear to me and appears to be split across 2 paragraphs. Please can this be reworded? Does some of the lines 135-138 paragraph refer to the whole mitochondrial genome sequencing methodology?

We apologize to the reviewer. We revised the indicated paragraph.

    Results, page 8, line 297: figure 3b should be referred to in the text here (as well as figure 3c).

We thank the reviewer for his suggestion. We revised our manuscript accordingly.

    Results, page 8, figure 3b: 2 lanes are marked as negative controls and appear to have different sized fragments; this should be corrected. Also, I think the positive control represents a positive control for multiple deletions; this should be clearly indicated (as has been done for figure 3c).

We thank the reviewer for his suggestion. We revised our manuscript accordingly.

    Discussion, page 10, line 372: “muscle biopsy in our subject 3 resulted only mildly affected” does not read well. It would be clearer to say “muscle biopsy in our subject 3 was only mildly abnormal.”

We thank the reviewer for his suggestion. We revised our manuscript accordingly.

    Discussion, page 10, lines 375-376: Again, this sentence is not clear. I suggest rewording along the following lines: “Therefore, it would not be surprising in subject 3, who has prominent involvement of respiratory muscles, to detect significant mitochondrial abnormalities in respiratory muscle tissue.”

We thank the reviewer for his suggestion. We revised our manuscript accordingly.

    Discussion, page 12, line 436: “TWKN” should be “TWNK”.

We thank the reviewer for his suggestion. We revised our manuscript accordingly.

Reviewer 3 Report

Comments and Suggestions for Authors

The manuscript by Diego Lopergolo and colleagues has been significantly improved and I agree with the reply, which the authors provided. Still, I have some concern because of the limited number of affected individuals, which is especially meaningful in combination with incomplete penetrance. Since the Southern blots from muscle DNA show the defect of mitochondrial DNA quite convincingly, I agree that this manuscript should be published. However, there is still a certain degree of uncertainty, which should be visualized by formulating more carfully:

  1. I would advise to change the title as follows:

“The first heterozygous TWNK nonsense mutation associated with progressive external ophthalmoplegia: >>evidence for<< a new piece in the puzzle of mitochondrial diseases”

  1. In the abstract change:

“Our findings confirm the high intrafamilial …”

to

“Our findings are >>in line with<< the high intrafamilial …”

  1. In the discussion (line 335) change:

“We confirm the high intrafamilial …”

to

“We >>observed a<< high intrafamilial …”

  1. In the conclusions accordingly to the abstract change:

“Our findings confirm the high intrafamilial …”

to

“Our findings are >>in line with<< the high intrafamilial …”

Minor points:

Line 126: “and >>S<<outhern blotting” in capital letters

Lines 273, 275, 433, and 436: “TWNK” in italics

Author Response

REVIEWER 3

The manuscript by Diego Lopergolo and colleagues has been significantly improved and I agree with the reply, which the authors provided. Still, I have some concern because of the limited number of affected individuals, which is especially meaningful in combination with incomplete penetrance. Since the Southern blots from muscle DNA show the defect of mitochondrial DNA quite convincingly, I agree that this manuscript should be published. However, there is still a certain degree of uncertainty, which should be visualized by formulating more carfully:

  1. I would advise to change the title as follows:

“The first heterozygous TWNK nonsense mutation associated with progressive external ophthalmoplegia: >>evidence for<< a new piece in the puzzle of mitochondrial diseases”

We thank the reviewer for his suggestion. We revised our manuscript accordingly.

  1. In the abstract change:

 “Our findings confirm the high intrafamilial …”

to

“Our findings are >>in line with<< the high intrafamilial …”

We thank the reviewer for his suggestion. We revised our manuscript accordingly.

  1. In the discussion (line 335) change:

“We confirm the high intrafamilial …”

to

“We >>observed a<< high intrafamilial …”

We thank the reviewer for his suggestion. We revised our manuscript accordingly.

  1. In the conclusions accordingly to the abstract change:

“Our findings confirm the high intrafamilial …”

to

“Our findings are >>in line with<< the high intrafamilial …”

We thank the reviewer for his suggestion. We revised our manuscript accordingly.

Minor points:

Line 126: “and >>S<<outhern blotting” in capital letters

We thank the reviewer for his suggestion. We revised our manuscript accordingly.

Lines 273, 275, 433, and 436: “TWNK” in italics

We thank the reviewer for his suggestion. We revised our manuscript accordingly.